# Considerations Concerning the Little Group

**Jens Erler**

PRISMA$^+$ Cluster of Excellence & Institute for Nuclear Physics, Johannes Gutenberg University, 55099 Mainz, Germany; erler@uni-mainz.de

**Abstract:** I very briefly review both the historical and constructive approaches to relativistic quantum mechanics and relativistic quantum field theory, including remarks on the possibility of a non-vanishing photon mass, as well as a foolhardy speculation regarding dark matter.

**Keywords:** quantum fields; continuous spin representations; photon mass; dark matter

## 1. Relativistic Quantum Mechanics

Historically[1], relativistic quantum mechanics ("first quantization") was simply understood as the relativistic generalization of the Schrödinger equation [2]. This is amusing, as Schrödinger apparently conceived of the relativistic wave equation for a free and spinless particle,

$$(\partial_\mu \partial^\mu + m^2)\phi = 0, \tag{1}$$

before Klein [3] and Gordon [4] did so, whose names are associated with it today. Including the electromagnetic interaction into Equation (1) fails to produce the correct fine structure of the hydrogen atom, so Schrödinger worked in the non-relativistic limit until after Refs. [3,4] had been published. Dirac was dissatisfied with Equation (1), as it seemed to lead to negative probabilities. His celebrated relativistic wave equation [5] for a free particle of spin 1/2,

$$(i\slashed{\partial} - m)\psi = 0, \tag{2}$$

supplemented with the electromagnetic interaction, was very successful in correctly predicting the magnetic moment of the electron, the existence of the positron, and the fine structure of hydrogen.

The Dirac theory had several drawbacks. It interpreted positrons as holes in an otherwise filled infinite sea of negative energy electrons, invoking the Pauli exclusion principle, which is not generalizable to bosons. This means, e.g., that the treatments of charged pions or the $W^\pm$ bosons remain obscure. Furthermore, Equation (2) cannot be applied to neutral scalar bosons such as the helium nucleus or the Higgs boson. It should also be noted that nothing prevents one to introduce a so-called Pauli term [6] with an arbitrary adjustable coefficient, which has the exact form of a magnetic moment and would render the prediction from the Dirac theory an accident.

In any case, Dirac constructed his theory for the wrong reason, thinking that the alleged negative probabilities of the Klein–Gordon theory were the deeper reason why the prominent matter fields of the time, electrons, and protons had to have spin 1/2, as otherwise, a consistent quantum theory seemed impossible. Today, it is understood that Equation (2) represents the correct relativistic equation for spin 1/2 particles, but other values for the spin are perfectly possible. The negative probabilities enter only when $\phi$ in Equation (1) is interpreted as a

probability amplitude rather than a quantum operator acting on a Hilbert space endowed with positive definite probabilities by construction.

The modern approach seeks to *construct* a theory uniting the foundations of quantum mechanics [7] with the principle of special relativity [8] from the start. The first step is to classify the unitary representations [9] of the Poincaré (inhomogeneous Lorentz) group [10], resulting in four distinct types:

**Massive particles.** The one-particle unitary irreducible representations (irreps) with $m^2 > 0$ mediate short-range interactions. To find additional quantum numbers, one chooses the rest frame of the particle, where the momentum four-vector has the form $P^\mu = (m, 0, 0, 0)$. The subgroup of Lorentz transformations which leave $P^\mu$ unchanged (the little group [9]) is the group $SO(3)$ of three-dimensional spatial rotations, so the familiar integer and half-integer spin representations of non-relativistic quantum mechanics are recovered.

**Massless particles.** The irreps with $m^2 = 0$ mediate long-range interactions. To find additional quantum numbers, one may choose a reference frame in which the momentum four-vector takes the form $P^\mu = (k, 0, 0, k)$. One can see that the group $SO(2)$ of two-dimensional spatial rotations (in the plane orthogonal to the $z$-direction) leaves the reference vector $P^\mu$ unchanged. This gives rise to the concept of helicity $h$ (the eigenvalue of the third component of the angular momentum operator) as an additional quantum number[2]. However, unlike in the case of $SO(3)$, there is no algebraic obstruction to allow arbitrary (real) values for $h$, and one needs to study the topological structure of the Poincaré group [1]. This involves the computation of its first homotopy group, and one finds that only integer or half-integer values of $h$ are allowed. On the other hand, in more than four spacetime dimensions, it suffices to consider infinitesimal Lorentz transformations, i.e., to classify the irreps of the underlying Lie algebra of symmetry generators, where one again finds integer and half-integer spin representations.

**Continuous spin particles.** The complete little group for $m^2 = 0$ states is the Euclidian group $ISO(2)$, extending the aforementioned $SO(2)$ to include two-dimensional translations. The latter are those combinations of Lorentz boosts and three-dimensional rotations that leave the reference momentum four-vector unchanged. However, the group $ISO(2)$ is neither compact nor semi-simple, and any representation for which the translations act non-trivially will possess a continuous spin variable. Such representations have not been observed in nature[3] and not much is known about them, but there are relatively recent attempts to construct interacting field theories for them [14]. The only property that is certain is that continuous spin particles would necessarily interact gravitationally. For quite some time, I have been wondering whether they may, in fact, play a role within the dark matter sector or, conversely, whether there is a way to exclude this possibility. However, as continuous spin particles are massless, they themselves could constitute dark matter candidates only if they can form massive bound states, loosely analogous to glueballs. Indeed, glueballs are known as a prime example of a self-interacting dark matter candidate [15]. Ref. [16] points to novel phenomena in the dark sector, especially for glueballs from a hidden $SU(N)$ gauge interaction with large $N$, including warm dark matter scenarios, Bose–Einstein condensation leading to supermassive dark stars, and indirect detections through higher dimensional operators, as well as interesting collider signatures. The relic abundance of dark glueballs has been studied in Ref. [17] in a thermal effective theory accounting for strong-coupling dynamics.

**Tachyons.** The little group for irreps with $m^2 < 0$ is $SO(1, 2)$, which, as a simple but non-compact group, permits only the trivial and infinite-dimensional representations. The appearance of tachyonic representations usually signals the presence of instabilities in a field theory, and may trigger the process of tachyon condensation, a phenomenon closely

related to a second-order phase transition. It may be possible to remove the tachyons by field re-definitions, as exemplified by the Higgs mechanism [18,19].

## 2. Quantum Fields

The first relativistic quantum field theories were constructed by quantizing a given classical field theory ("second quantization"), specifically the Maxwell equations [20]. This was a complicated and confusing process, particularly due to the presence of ultra-violet divergencies [21,22]. The approach was ultimately successful, but only after the protagonists [23,24] had almost given up or were exploring alternatives [25].

Now, the Wigner classification [9], summarized in Section 1, is only a necessary step to construct a relativistic quantum theory from first principles. One also needs a unitary (probability conserving) and Lorentz-invariant (relativistic) scattering matrix (*S*-matrix) for the interacting theory. The more recent and very successful developments keep Lorentz covariance manifest, computing helicity amplitudes at the tree [26,27] and loop [28] levels. Unitarity, which is not manifest in this approach, is used as a tool instead.

Conversely, the traditional formulation of quantum field theory keeps unitarity manifest. One builds Hamiltonian densities $\mathcal{H}$ out of sums of products of annihilation ($a$) and creation ($a^\dagger$) operators for particles of specific momentum $\vec{p}$, spin (or helicity), and type[4]. On the other hand, Lorentz covariance of the *S*-matrix is implemented by imposing a (sufficient) causality condition, usually in the form,

$$[\mathcal{H}(x), \mathcal{H}(y)] = 0 \qquad (x - y)^2 \le 0. \tag{3}$$

Equation (3) is a condition in configuration space where the one-particle states and the corresponding $a^\dagger$ that create them from the vacuum have definite momentum. Thus, to construct $\mathcal{H}$, we need to introduce quantum fields as the Fourier transforms of the $a$ and $a^\dagger$ operators. Since the latter have definite Lorentz transformation properties, one can *calculate* the Lorentz transformation rules of the fields, in contrast to classical field theories where the Lorentz transformation rule is *postulated*. For example, spin 0 particles are represented by Klein–Gordon fields, spin 1/2 particles by Dirac fields, and massive spin 1 particles by Proca fields [29]. These fields turn out to have covariant Lorentz transformations, so Lorentz-invariant theories can be constructed from them straightforwardly.

However, if $a$ and $a^\dagger$ are the annihilation and creation operators for a massless particle with helicities $h = \pm 1$, then gauge fields such as the photon would be of the form,

$$A_\mu(x) = \int \frac{d^3 p}{(2\pi)^3 2E_{\vec{p}}} \sum_{h=\pm 1} \left[ \epsilon_\mu(\vec{p}, h) e^{-ipx} a(\vec{p}, h) + \epsilon_\mu^*(\vec{p}, h) e^{ipx} a^\dagger(\vec{p}, h) \right], \tag{4}$$

where $\epsilon_\mu$ and $\epsilon_\mu^*$ are polarization vectors and $E_{\vec{p}}$ denotes the energy. The transformation behaviour of $A_\mu$ turns out to be:

$$U(\Lambda) A_\mu(x) U(\Lambda)^{-1} = \Lambda^\nu{}_\mu A_\nu(\Lambda x) + \partial_\mu \Omega(\Lambda, x), \tag{5}$$

where $\Lambda^\nu{}_\mu$ is the Lorentz transformation matrix and $\Omega$ a linear combination of $a$ and $a^\dagger$. The last term shows that $A_\mu$ does not transform as a Lorentz vector. One could utilize a different kind of field, such as an antisymmetric two-index tensor field $B_{\mu\nu}$, but this will not give rise to long-range interactions. The way out is to construct $\mathcal{H}$ in such a way that it is invariant under a gauge transformation,

$$A_\mu(x) \to A_\mu(x) + \partial_\mu \Omega(x), \tag{6}$$

so that $\mathcal{H}$ is invariant under a Lorentz transformation followed by a gauge transformation. This implies that breaking gauge invariance is tantamount to breaking Lorentz invariance.

A similar line of arguments shows that the equivalence principle arises when one attempts to construct a theory for a massless particle of helicity $h = \pm 2$ with long-range interactions. Thus, neither the gauge nor the equivalence principle need to be postulated in the quantum context (only the very existence of the massless $h = \pm 1$ or $h = \pm 2$ particles with long-range interactions).

### 3. Stückelberg Mechanism

In a gauge invariant theory, the gauge fields can only enter through gauge covariant derivatives or the field strength tensor, whereas an explicit appearance would be incompatible with Equation (6). In particular, mass terms violate gauge invariance, explaining why gluons are massless. However, for an Abelian theory such as QED, there is a loophole [30,31] (it is absent for theories based on non-Abelian gauge groups such as QCD or general relativity; see, e.g., Ref. [32]). Consider the Lagrangian,

$$\mathcal{L} = -\frac{1}{4}(\partial_\mu A_\nu - \partial_\nu A_\mu)^2 + \frac{1}{2}(m A_\mu - \partial_\mu B)^2 - \frac{1}{2}(\partial_\mu A^\mu + m B)^2, \tag{7}$$

where $B(x)$ is a real scalar field. The first two terms in $\mathcal{L}$ are invariant under the gauge transformation [6],

$$A_\mu(x) \to A_\mu(x) + \partial_\mu \Omega(x) \qquad\qquad B(x) \to B(x) + m\Omega(x) \tag{8}$$

whereas the third term is invariant, provided that the gauge parameter $\Omega(x)$ satisfies the Klein–Gordon equation,

$$(\partial_\mu \partial^\mu + m^2)\Omega(x) = 0. \tag{9}$$

A restriction such as (9) is absent in QED, but the important point is that it does not change the number of degrees of freedom. The second term in $\mathcal{L}$ gives rise to a mass term for $A_\mu$ and the kinetic term for $B$.

If one now defines the gauge-invariant combination [6],

$$V_\mu(x) \equiv A_\mu(x) - \frac{1}{m}\partial_\mu B(x), \tag{10}$$

one obtains:

$$\mathcal{L} = -\frac{1}{4}(\partial_\mu V_\nu - \partial_\nu V_\mu)^2 + \frac{m^2}{2}V_\mu V^\mu - \frac{1}{2}(\partial_\mu A^\mu + m B)^2, \tag{11}$$

so one is left with a massive vector boson $V_\mu(x)$ in a theory with the additional constraint, so that matrix elements of $\partial_\mu A^\mu + m B$ taken between physical states must vanish. This is reminiscent of QED where one needs to demand the vanishing of the matrix elements of $\partial_\mu A^\mu$ taken between physical states. Effectively, we have traded the degree of freedom represented by $B(x)$ for the longitudinal polarization (the helicity $h = 0$ state) of a massive vector boson $V_\mu(x)$, since the first two terms in Equation (11) coincide with the Lagrangian of the Proca field [29] for a massive spin 1 particle. The difference is that one has $\partial_\mu V^\mu = 0$ following directly from the Euler–Lagrange equations.

The fact that the Stückelberg mass derives from a gauge-invariant Lagrangian may be used to show that this theory is perturbatively renormalizable. Indeed, the last term in Equation (11) can be interpreted as a gauge-fixing term, corresponding in the case $m = 0$ (QED) to the Feynman gauge. For a clear discussion, see Ref. [33].

## 4. Conclusions

Following the line of thought reviewed in this note, we conclude that there are three different ways to obtain a mass for the photon.

- One can speculate that Lorentz invariance may not be an exact symmetry of Nature [34]. Due to the intimate connection between Lorentz and gauge invariance, one may expect that Lorentz invariance will protect the photon from acquiring a mass only up to (large) length scales at which it is broken. Ref. [35] applies this idea to a specific scenario involving both Lorentz symmetry and supersymmetry breaking, and Ref. [36] discusses the effects of a hypothetical photon mass from Lorentz symmetry breaking in the context of standard cold dark matter cosmology with a cosmological constant ($\Lambda$CDM), challenging the paradigm of a universe with an accelerating expansion. Radiative corrections in a vector model with spontaneous Lorentz symmetry violation have also been addressed [37].
- The Stückelberg mechanism adds an additional degree of freedom to the theory that in a particular gauge may be interpretable as the longitudinal polarisation (in addition to the two transversal ones) of a massive vector boson, as in the case of a Proca field. There is an interesting argument [38] (resting on conjectures) that in the context of gravity, the very small photon masses ($m_\gamma \lesssim 10^{-18}$ eV [39]), consistent with observations[5], would imply an ultraviolet cutoff that is too low. A Stückelberg photon mass would then be ruled out.
- One can also employ the Higgs mechanism [18,19], in which case the longitudinal degree of freedom of the photon is provided together with an additional physical scalar degree of freedom in complete analogy with the masses of the *W* and *Z* bosons in the standard model [42]. However, one would face an aggravated hierarchy problem, as the small photon mass would generally not be protected to be driven to very large masses by radiative corrections.

The most likely way the photon may be tied to the dark matter sector would be through its mixing with a dark massive photon [43]. Indeed, kinetic mixing would occur at the renormalizable level, i.e., in principle unsuppressed by large new physics scales. A variant of this scenario adds mass mixing of the dark photon with the standard model *Z* boson [44]. Unlike a dark photon, such a dark Z boson could induce observable effects in atomic parity violation experiments [45] and parity violating electron scattering [46].

It will also be interesting to study whether the hypothetical continuous spin particles could exist in nature, and if so, how they would couple to the standard model sector other than gravitationally. Additionally, one would need to find out whether they would play a rôle in dark sector physics[6].

**Funding:** This research received no external funding.

**Data Availability Statement:** Data sharing not applicable.

**Acknowledgments:** It is a pleasure to thank Dmitry Budker, Rodolfo Ferro, Misha Gorshteyn, Dmitri Ryutov, Hubert Spiesberger, and Arne Wickenbrock for reading this paper with great attention to detail and for their many valuable comments and suggestions.

**Conflicts of Interest:** The author declares no conflict of interest.

## Notes

[1]   Most of the remarks on the historical development of the relativistic quantum theory that follow are inspired by Ref. [1].

[2]   The CPT theorem [11,12] implies that helicity states come in pairs, i.e., if the helicity $h$ of a particle is present, then so is a state with helicity $-h$ and equal, but opposite, quantum numbers. For example, the single degree of freedom represented by a right-handed circularly polarised photon ($h = +1$) is by itself an irrep, as no Lorentz transformation can change the polarisation of a massless particle. Nevertheless, in a consistent relativistic quantum theory, left-handed circularly polarised photons ($h = -1$) need to be included as well. Likewise, a massless left-handed neutrino ($h = -1/2$) will always be accompanied by a right-handed anti-neutrino ($h = +1/2$).

[3]   In condensed matter systems, they may be understood as massless generalisations of anyons [13].

[4]   This represents no loss of generality, as any operator may be expressed in this way.

[5]   For a review on photon and graviton mass limits, see Ref. [40]. The effect of a finite photon mass on galactic rotation curves through the modification of plasma electrodynamics has been considered in Ref. [41].

[6]   While this manuscript was under review, a paper [47] appeared presenting Feynman rules for computing scattering amplitudes involving the exchange of continuous spin particles. This work may open the way to more quantitatively address the effects of these kinds of exotic states, including their possible relevance for dark matter.

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
