# Peer review of "Considerations Concerning the Little Group"

_universe, doi:10.3390/universe9090420_

Round 1
Reviewer 1 Report
The historical and constructive approaches to relativistic quantum mechanics and relativistic quantum field theory are discussed in this overview, along with some naive speculation about dark matter and the potential of a non-vanishing photon mass. In the setting of conventional Cold Dark Matter cosmology with a cosmological constant (ΛCDM), the effects of a hypothetical photon mass from Lorentz symmetry breaking are examined. It is also emphasized how the Stückelberg mechanism offers a further degree of freedom to the theory that, depending on the gauge, may be interpreted as the longitudinal polarization of a massive vector boson. In the Higgs mechanism, which is completely analogous to the masses of the W and Z bosons in the Standard Model, the photon's longitudinal degree of freedom is provided along with an additional physical scalar degree of freedom. However, this simplistic piece is too brief to serve as a review. You should elaborate on each point.
Author Response
Thank you for your careful reading of the manuscript. In the revised version I have added some remarks regarding the naive speculation about dark matter, where continuous spin representation (CSR) might form bound states in analogy to glueballs. Here, I added and described three references [16,17,18], where [16] is a more general reference introducing the idea of self-interacting dark matter, [17| points to several phenomenological aspects, and [18] is on the relic density (of glue ball dark matter). As for the CSR themselves, it is difficult to be more quantitative at this point, as the field theory describing these states is still being developed. However, I added a note on a very interesting paper [48] that just appeared with important progress in this respect. I also changed the title from "Perspective on the Little Group" to "Considerations concerning the Little Group" and have the article type re-classified as opinion rather than perspective.
Reviewer 2 Report
The article revisits the plausible ways of achieving massive photons and it's possible linkage to Dark Matter in the form of Dark photon. The articles discusses these issues in brief and articulate way. I would recommend the publication of this article in the present form.
Author Response
Thank for reading through the manuscript and your comments
Reviewer 3 Report
The review shows various ways to add mass to a vector field.
It remains unclear how this vector field could fulfil the role of a dark matter.
Author Response
Thank for reading through the manuscript and your comments.